# Agglomeration costs limit sustainable innovation in cities in developing economies

**Saul Estrin**[1]*, **Yuan Hu**[2], **Daniel Shapiro**[3], **Peng Zhang**[3]

**1** Department of Management, London School of Economics and Political Science, London, United Kingdom,
**2** Department of Economics and Public Policy, Imperial College Business School, London, United Kingdom,
**3** Beedie School of Business, Simon Fraser University, Burnaby, Canada

☯ These authors contributed equally to this work.
\* s.estrin@lse.ac.uk

**Data Availability Statement:** Data from the WBES are freely available. Users only need to register with the Enterprise Analysis Unit at the World Bank by completing the Enterprise Surveys Data Access Protocol (https://login.enterprisesurveys.org/content/sites/financeandprivatesector/en/signup.

## Abstract

Theory and evidence from developed economies suggests that innovation activities benefit from agglomeration economies associated with urban economic density. However, despite the fact that eighteen of the world's top twenty cities are in developing countries, we do not know whether agglomeration affects innovation in the same way in developing countries. We propose that, while there are still agglomeration benefits, the development path followed by cities in developing countries also creates significant agglomeration costs and these act to limit innovation. We build a unique database to measure consistently both urban economic density and innovation across a large number of developing countries. Based on geospatial information, we combine data on nightlights at the city level to proxy urban density with information on innovation activity at the firm level. We find that in developing countries, as urban economic density increases, innovation first increases and then begins to decrease beyond a certain point, with the decline being most prominent in the largest cities. That is, the largest cities in developing countries are not able to act as sustainable sources of innovation. Cities in developing countries therefore display different patterns of agglomeration from those documented in the literature focused on developed countries. Our analysis explores the relationship between UN Sustainable Development Goal (SDG) 9 which fosters innovation, and SDG 11 which promotes sustainable and resilient cities. Our results suggest the importance of addressing urban agglomeration costs as a means to facilitate innovative activity.

## Introduction

Considerable cross-disciplinary literature and significant empirical evidence from developed economies shows that innovative activities are concentrated in large cities [1–6]. Cities are argued to provide *agglomeration benefits* in the form of: a) positive externalities via knowledge spillovers resulting from the enhanced economic and social interactions associated with greater urban density [7–11]; and b) economies of scale [12–17] drawing on suitable provision of infrastructure [18]. These agglomeration benefits enable the aggregation and co-location of

html). There are two download options: a) data by economy; b) combined data. We use "combined data" because it has a standard set of questions in different countries, which is suitable for cross country comparisons. Data on (geo masked) GPS coordinates for each firm is available since 2010. GPS data is also accessible for free, but a research proposal is required for approval. Harmonized nightlight data is publicly available from: https://doi. org/10.6084/m9.figshare.9828827.v2. There is no restriction on the availability of nightlight data.

**Funding:** Peng Zhang acknowledges financial support from the Social Sciences & Humanities Research Council of Canada (Project Number: 430-2021-00559): https://www.sshrc-crsh.gc.ca/home-accueil-eng.aspx. The funder did not play any role in the study design, data collection, analysis or preparation of the manuscript.

**Competing interests:** The authors have declared that no competing interests exist.

complex knowledge-based activities. It has been proposed that the processes linking urban agglomeration and innovation are shared by cities across different nations, urban systems, and times [2, 19].

However, there are also reasons to question whether these theories apply to cities in *developing* countries [18, 20, 21]. This is because cities, especially very large cities, in developing countries may follow different processes of agglomeration [22–27], which can lead to sustainable development challenges including less innovation. Thus, cities in developed countries typically developed gradually and were distributed relatively evenly across geographies because historically high transport costs between regions restricted agglomeration. As a result, urban infrastructure was able to keep pace with or catch up with population growth. However, developing economies have mostly urbanised more recently, when transport costs were lower, and much more rapidly. As a result, while population and economic activities have quickly become concentrated in relatively few locations (eighteen of the world's top twenty cities by population are in developing countries [28]), the development of infrastructure has often not yet kept pace [23], leading to inadequate infrastructure and congestion. This may hinder knowledge spillovers and reduce innovative activities. Although the potential costs of this high and rapid agglomeration in developing countries have been recognised, they have not yet been systematically analysed, especially with respect to innovation [26, 29–36].

Such empirical evidence as there is on agglomeration effects in developing countries that takes into account agglomeration costs is often based on individual or small sets of countries and considers outcomes other than innovation [26, 29, 30, 36, 37]. Studying the impact of urban agglomeration on innovation in a large cross-sectional context is therefore viewed as an urgent research task. This task is challenging due to the "absence of comprehensive, consistently-defined, and reliably collected data on urban economic output" [38].

In order to meet these challenges, this paper analyses the relationship between urban agglomeration and innovation in a broad sample of developing countries. Furthermore, we use measures that are consistently defined across a large set of countries. We pay special attention to the large cities that characterise the developing world, and we base our empirical analysis on the possibility that agglomeration costs may limit innovation in developing countries.

Our study makes two main contributions. First, we provide a framework conceptualising the possibility that urban agglomeration does not always improve innovation performance in developing countries: indeed, it may hinder it. We propose that while cities in developing countries may still provide agglomeration benefits for firms that innovate, agglomeration costs may be large enough to offset those benefits when urban economic density is high, thus reducing innovation. This suggests that the relationship between innovation and urban agglomeration in developing countries will be *concave* and hill-shaped with a pronounced declining segment. Consistent with the argument that high agglomeration costs in cities in developing countries may be linked to rapid urbanisation resulting from their particular development path, we further propose that the declining part of the curve is mostly driven by cities with the largest populations. These cities face significant problems of disease, crime, congestion and high costs of travel and communication which may limit economies of scale [26]. It is an important finding that agglomeration costs hinder the innovation potential of large cities, as these cities might otherwise be innovation hubs in the developing world.

Our second contribution tackles the challenge faced by empirical studies on a global scale in developing countries: finding comprehensive, consistently defined measures and reliably collected data on both urban agglomeration and innovation. Recent advances in data collection allow us to create a unique data base which measures both innovation and urban agglomeration in a consistent fashion across a large sample of developing countries.

We study innovation activity at the firm level, thereby responding to calls for more micro-level analysis when studying urban scaling [2, 35, 39]. We measure firm-level innovation activity in a consistent way across countries by using World Bank Enterprise Survey (WBES) data, derived from a standard survey instrument applied consistently in a large number of developing economies (https://www.enterprisesurveys.org/en/data). Urban agglomeration is measured as urban economic density, which is proxied by nightlight (NTL) density at the city level. Recent research suggests that NTL density is a good proxy for the density of urban economic activities [23, 40], especially in developing economies given the deficiencies of more traditional data sources [38]. NTL density can be understood to measure the socio-economic complexity at the heart of urban density, therefore getting at the root of agglomeration effects [38, 41, 42]. Both NTL intensity, constructed for our study at the city level across countries and time, and the WBES innovation data at the firm level, are therefore measured consistently and reliably across our sample, thus overcoming a major obstacle in empirical studies across a large set of cities and countries in the developing world. Details are found in the Methods section.

The World Bank includes geo-positioning data (GPS coordinates of each surveyed firm) since 2011 which allows us to match NTL luminosity data with WBES firm-level data based on cities and survey years. Our sample contains 164 unique cities from 96 countries containing some 33,000 firms (the number can vary with the measure of innovation) and representing all major economic sectors. As detailed in the Methods section, in building our sample of country-cities, we control for cross-country variations in city systems by selecting fewer (more) cities in smaller (larger) countries. Characteristics of the sample are illustrated in Fig 1.

On this basis, we estimate equations to link firm level innovation and NTL density in the city in which the firm is located. Our dependent variable is an innovation index which we construct from the WBES data, that indicates whether the firm introduced a) a new product or service (new product); b) a new process (new process); c) engaged in R&D spending (R&D). The index is the sum of the individual components, a+b+c. Our independent variable is NTL density, measured as the sum of NTL luminosity within a city territory divided by the total area of the city. We take the logarithm of this NTL density measure and lag it by one year to mitigate potential reverse causality issues. We test our concavity proposition using regression models. We estimate a quadratic polynomial function of NTL intensity on firm innovation in which the linear and superlinear functions are nested. We reject both linearity and superlinearity in favour of our proposed concave function with a pronounced declining segment [43, 44], as discussed in the Results section.

At lower levels of NTL density, we find that firm level innovation is higher across the sample as cities become larger. However, firms' innovation levels fall as NTL density increases beyond a certain level. This is consistent with the interpretation that agglomeration benefits do exist, but agglomeration costs in developing countries rise with urban density [9]; a phenomenon not typically found in studies based on developed countries where the relevant social and physical infrastructure accumulates over time to limit or prevent such constraints from binding. Moreover, our results not only are different from those found in the studies of developed countries today, but also may be different from studies on now-developed countries in the past, a point related to but different from existing literature [45, 46].

Our findings have important policy implications. We address the issues of sustainable cities and innovation by linking two Sustainable Development Goals (SDGs) adopted by the United Nations as part of the 2030 Agenda (SDG 9 on innovation and SDG 11 on sustainable cities). Our results imply that high urban economic density sits at the root of key challenges to sustainable cities notably the largest cities. Therefore, balancing the benefits and costs of urban agglomeration to enhance innovation is a timely and urgent task. Our findings also pose

Panel a

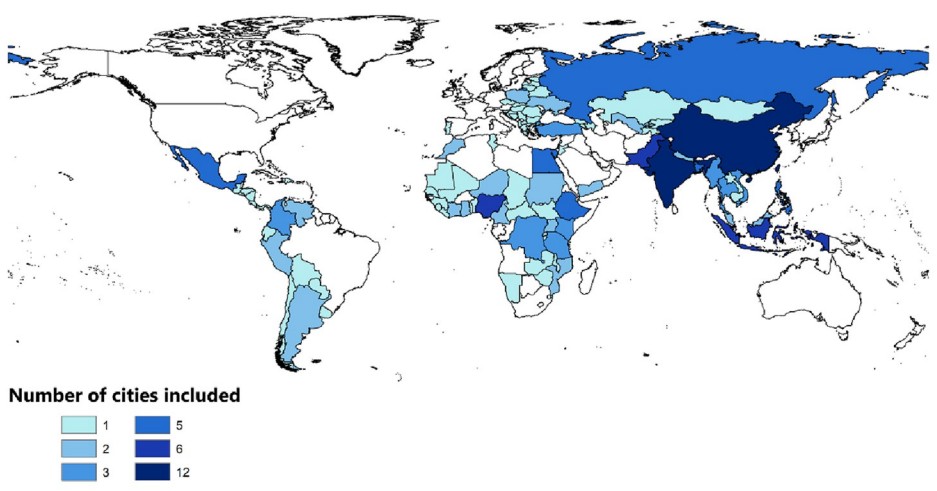

**Number of cities included**

1   5
2   6
3   12

Panel b

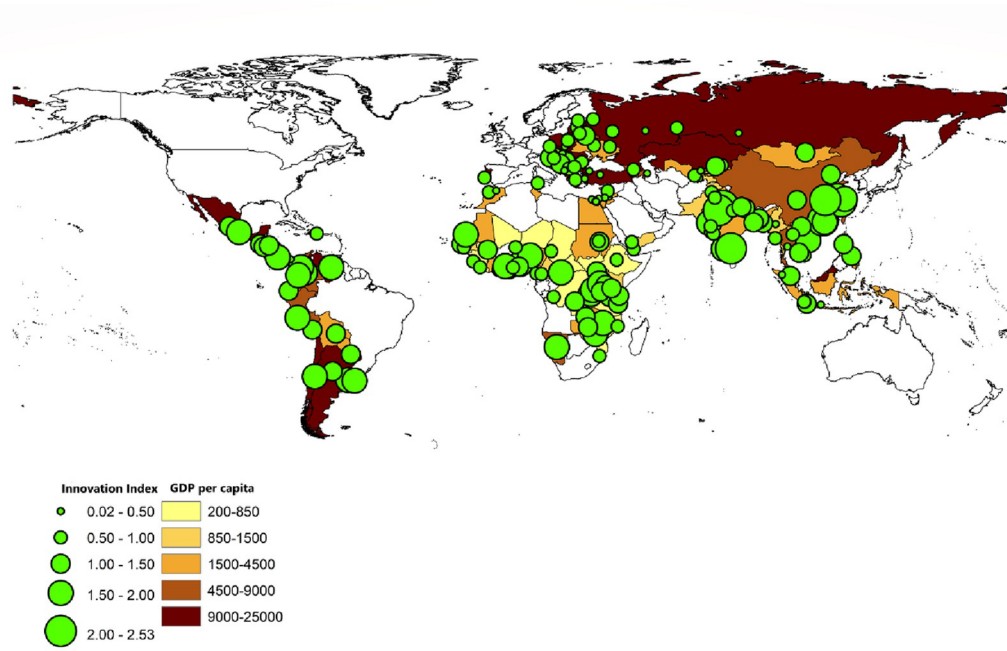

Innovation Index   GDP per capita
0.02 - 0.50   200-850
0.50 - 1.00   850-1500
1.00 - 1.50   1500-4500
1.50 - 2.00   4500-9000
2.00 - 2.53   9000-25000

**Fig 1. Sample characteristics: Countries, cities and city innovation.** Panel **a** indicates the countries included in the sample, and the number of cities in each country. The final sample contains 96 countries and 164 cities. Inclusion criteria are explained in the Methods section. Panel **b** highlights the cities in the sample, the GDP per capita of the sample countries, and the aggregate innovation score of each city. Our regression analysis is at the firm level, and we do not use city aggregates; in this figure they are used for illustration. The innovation index is derived from the WBES data as

explained in the Methods section. For cities that appear twice in the WBES data, the average NTL and/or innovation values for those two years were employed. Maps of city boundaries come from the Database of Global Administrative Areas (https://gadm.org/download_country_v3.html). We use ArcGIS to match GPS coordinates of firms with city locations. Source: own calculations.

urgent questions for policymakers in developing countries because the innovation benefits foregone are most pronounced in precisely those large cities which could potentially play the leading role in innovation [47, 48]. This points to the role of public policies in large cities in developing economies to address the costs of excessive agglomeration and the negative externalities they create.

## Results

Use of an index of innovation [49] mitigates the possible biases resulting from the use of single measures of innovation [3]. The three individual measures each take the value unity or zero (yes or no), so the innovation index is a count between 0 and 3. Firms are considered to have no, low, medium and high levels of innovation if this innovation index equals 0, 1, 2 and 3 respectively. Table 1 reports that the proportions of firms in each category are large enough to generate valid statistical inference: 42.7% (s.d. = 0.495) of firms have no innovation, 22.2% (s.d. = 0.415) have a low level of innovation, 21.4% (s.d. = 0.410) have a medium level of innovation and 13.7% (s.d. = 0.344) have a high level of innovation. These four numbers sum to 100%. The table also presents the summary statistics for individual items in the innovation index: 42.6% of firms developed a new product; 40.8% launched a new process and 22.8% reported engaging in R&D expenditures. The mean value of nightlight density by city is 38.921

**Table 1. Summary statistics of key variables.**

| Variables | Mean | Std | No. of obs |
|---|---|---|---|
| *Statistics for innovation index* | | | |
| No innovation (Index = 0) | 0.427 | 0.495 | 31798 |
| Low innovation (Index = 1) | 0.222 | 0.415 | 31798 |
| Medium innovation (Index = 2) | 0.214 | 0.410 | 31798 |
| High innovation (Index = 3) | 0.137 | 0.344 | 31798 |
| *Statistics for individual items* | | | |
| New Product | 0.426 | 0.494 | 31798 |
| New Process | 0.408 | 0.491 | 31798 |
| R&D | 0.228 | 0.419 | 31798 |
| *Statistics for night light data* | | | |
| Night Light Density | 38.921 | 16.475 | 31798 |

The summary statistics of innovation and nightlight density are shown in the table, which include mean values and standard deviations. In our sample, the proportions of firms in each innovation level are large enough to generate valid statistical inference: 42.7% of firms have no innovation, 22.2% have a low level of innovation, 21.4% have a medium level of innovation and 13.7% have a high level of innovation. These four numbers add up to 100%. The table also presents the summary statistics of individual items in the innovation index: 42.6% of firms developed a new product; 40.8% launched a new process and 22.8% had R&D expenses. The mean level of nightlight density is 38.921 (the range of night light luminosity is 0–63). The number of observations having non-missing data in **all** three items of the innovation index is 31,798. For each individual item, the number of observations with non-missing data is larger than 31,798, as reported in Table 2. The number of observations varies slightly depending on the measure of innovation, as explained in the Methods section.

**Table 2. Innovation and night light intensity for cities in developing countries.**

| VARIABLES | (1) | (2) | (3) | (4) |
|---|---|---|---|---|
| | **New Product** | **New Process** | **R&D** | **Innovation Index** |
| Ln(Night Light) | 0.155*** | 0.356*** | 0.169*** | 0.248*** |
| | (0.000) | (0.000) | (0.000) | (0.000) |
| | [0.086,0.225] | [0.284,0.428] | [0.082,0.256] | [0.187,0.309] |
| Ln(Night Light) Sqr | -0.022*** | -0.064*** | -0.020** | -0.039*** |
| | (0.004) | (0.000) | (0.034) | (0.000) |
| | [-0.037,-0.007] | [-0.080,-0.049] | [-0.038,-0.001] | [-0.052,-0.026] |
| Per Capita GDP | -0.048 | 0.009 | 0.287*** | 0.045 |
| | (0.223) | (0.835) | (0.000) | (0.196) |
| | [-0.125,0.029] | [-0.077,0.095] | [0.191,0.383] | [-0.023,0.114] |
| Constant | -1.259*** | -1.390*** | -1.394*** | |
| | (0.000) | (0.000) | (0.000) | |
| | [-1.482,-1.036] | [-1.625,-1.154] | [-1.673,-1.115] | |
| Observations | 32,675 | 32,095 | 32,171 | 31,798 |
| Conflict regions | Exclude | Exclude | Exclude | Exclude |
| GDPpc>30K countries | Exclude | Exclude | Exclude | Exclude |
| Region Fixed Effects | Yes | Yes | Yes | Yes |
| Year Fixed Effects | Yes | Yes | Yes | Yes |
| LR chi2(1) | 8.14 | 67.15 | 4.53 | 33.83 |
| Prob > chi2 | 0.004 | 0.000 | 0.033 | 0.000 |
| Pseudo R2 | 0.080 | 0.132 | 0.084 | 0.074 |

The table reports logit regressions for each individual item in the innovation index in columns 1–3, and the ordered logit regression for the innovation index in column 4. The key independent variables are the logarithms of nightlight density and its quadratic term. We control for per capita GDP in each country and include geographic region and year fixed effects. P-values are in parentheses, and 95% confidence intervals are in square brackets below p-values. Ordered Logit estimates do not include a constant. Number of observations varies because of missing values for each measure. NTL and GDP are lagged. Coefficients can be interpreted as the increase in the log odds of being in a higher innovation level versus all lower innovation levels after a 1% increase in nightlight density. For individual items, the POC is found to be a little lower for new processes (20.1), and higher for new products (54.6). For R&D (70.1), the POC occurs above the maximum value of NTL. The values are not directly comparable because the samples are somewhat different in each case.

\*\*\* p<0.01,

\*\* p<0.05,

\* p<0.1

(s.d. = 16.475; the range of night light luminosity is 0 to 63). Cities with the lowest nightlight luminosity include Juba (South Sudan) and Caracas (Venezuela). Cities with the highest night-light luminosity are those in the Nile Delta (Egypt) and Kolkata (India).

Our baseline regression results are reported in Table 2 where we present the estimated effects of (lagged) NTL on firm innovation in the city in which the firm is located, controlling for country-level per capita GDP and region and year fixed effects. For completeness, in columns 1–3, we present results for each of the three binary measures of innovation (new products, new processes, and R&D) using logit models. Column 4 reports the results for the innovation index based on an ordered/ordinal logit model. Each individual measure of innovation and the composite index generate consistent results. The ordered logit model captures the ordered categorical feature of the dependent variable and has advantages over both logit (which loses information by binarising the outcome) and OLS (which requires a strong assumption of linearity) models [50]. See S1 File and S7 Table for a detailed discussion, including a test of the assumptions for the ordered logit model.

Our baseline regression results in Table 2 form the basis of our conclusion that the underlying relationship between NTL intensity and innovation in developing economies is concave. Commencing with the central results, in column 4, the linear term of nightlight density is positively associated with the innovation index (b = 0.248, two-tail p<0.001, 95% CI = 0.187 to 0.309). However, the squared NTL term is negative and statistically significant (b = -0.039, two-tail p<0.001, 95% CI = -0.052 to -0.026), supporting the concavity of the function. The linear model is nested in the quadratic model and the squared NTL term is statistically significant. We reject the null hypothesis of linearity because the likelihood ratio tests show that models with the squared NTL term and the linear term fit the data significantly better than models with the linear term only: $\chi^2$ = 33.83 and p<0.001. Indeed, there is evidence that the relationship becomes negative in the relevant range, supporting the argument that agglomeration costs can outweigh the benefits of agglomeration for innovative activities in developing countries. Results on the individual innovation measures in columns 1–3 reveal similar patterns, further supporting the proposition of concavity and a pronounced declining part of the curve. Our results therefore indicate that a linear or superlinear relationship between innovation and urban density does not generalise outside the developed economy context, at least for the large sample of cities and countries we consider.

Our results show that, not only is the relationship concave to the origin, but that the maximum value of innovation is reached well within the observed range of luminosity. To the right of this maximum, which we term the *Point of Congestion* (PoC), increased NTL luminosity reduces firm-level innovative activity across cities. The PoC can be seen as describing the level of luminosity at which the marginal benefits of agglomeration for innovative activity equal the marginal costs of urban congestion. For the innovation index, the curve turns down when NTL is 22.760 (ln (NTL) = 3.125). 75% of the year-city observations in our sample are found on the declining part of the curve. Our results therefore suggest that the benefits of city agglomeration on innovation for developed economies are outweighed in developing countries by agglomeration costs in many or most major cities. Though agglomeration facilitates innovation in many cities in developed economies because of positive externalities and economies of scale in infrastructure development, such benefits may fail to materialise in developing countries in cities with high urban economic density because investments in public goods may not keep pace with rapid urbanization and population growth. The lack of adequate infrastructure may render a large city unable to maintain efficient or low-cost internal transport, logistics, health services or communications.

Agglomeration costs of the type noted above may be more prevalent in the largest cities. This suggests that the declining part of the innovation-NTL relationship may be driven by the cities with the largest populations. We therefore extend our analysis by adding an interaction term between city population size and NTL density in our ordered logit regression. At the same level of urban economic density, larger populations indicate more congestion and less sufficient public goods provision, leading to higher agglomeration costs. As logit coefficients are difficult to interpret directly with interactions, we follow the mainstream literature [51, 52] and present the effects of city size visually in Fig 2 by plotting the predicted probabilities of innovation outcomes (z-axis) across night light density (x-axis) on a continuous spectrum of city population size (y-axis) while controlling for the direct effects of NTL on innovation and holding all other variables at mean values. We consider two innovation outcomes in the z-axis: innovation intention (Fig 2a and 2b), calculated as the (predicted) cumulative probability of having low, medium and high levels of innovation (innovation index = 1, 2, or 3 versus 0); and innovation intensity (Fig 2c and 2d), calculated as the predicted probability of having a high level of innovation (innovation index = 3 versus others). We present the same 3D graph from

Panel a. Innovation intention, front | Panel b. Innovation intention, back

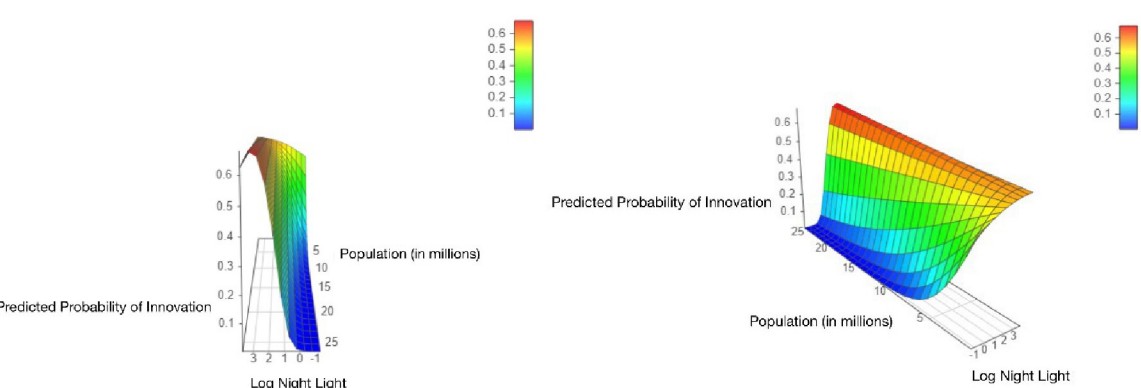

Panel c. Innovation intensity, front | Panel d. Innovation intensity, back

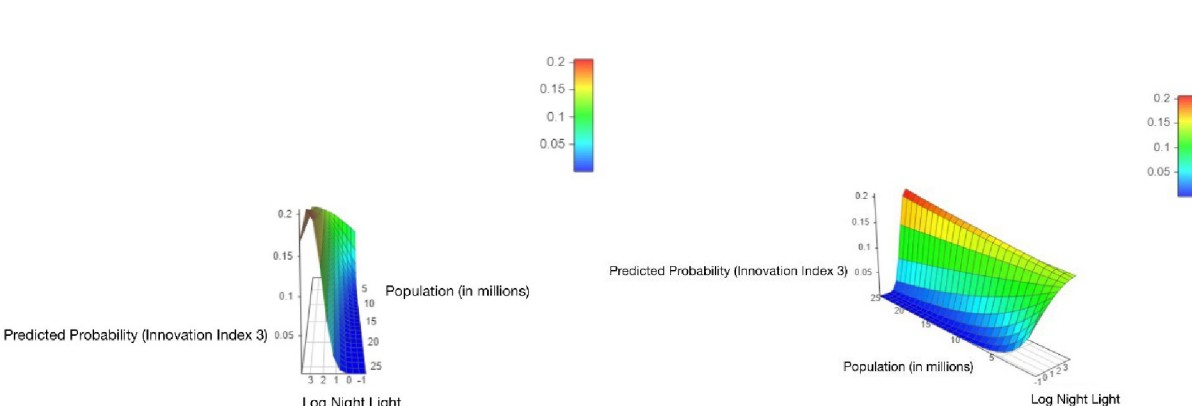

**Fig 2. Innovation, nightlights and a continuous measure of city size.** Figs **a-d** report the relationship between innovation and nightlights for the continuous spectrum of city population sizes. The graphs plot the predicted outcomes of innovation (z-axis) based on the interaction between nightlight intensity (x-axis) and city population (y-axis) using an ordered logit model. The outcome of innovation in **a** and **b** is innovation intention, calculated as the cumulative (predicted) probability of having low, medium and high levels of innovation. The outcome of innovation in **c** and **d** is innovation intensity, calculated as the predicted probability of having a high level of innovation. Figs **a** and **c** present the 3D graphs from the front angle and **b** and **c** present the same graphs from the back angle. Data on city population size is obtained from the most recent census data for each country, and only counts the population in the city proper or urban areas, whichever is consistent with the city boundary defined in our analysis. Darker color indicates higher night light luminosity. S3 Table reports the regressions upon which these 3D graphs are based.

different angles (front angle in **a** and **c**, back angle in **b** and **d**). S3 Table reports the regressions upon which these graphs are based.

The four graphs illustrate that compared with small cities, large cities have more innovation activities, though their innovation potential may be hindered by congestion effects. First, **a** and **c** show that the inverted-U shape between NTL density and innovation does not occur in cities with smaller populations, suggesting higher agglomeration costs likely be prevalent in larger cities. Second, in **b** and **d**, the maximum values of innovation in smaller cities are below the maximum values of innovation in larger cities, indicating that large cities are the innovation engines. These patterns are evident for both innovation intention and innovation intensity.

Taking this argument further, we split the sample to distinguish between smaller and larger cities with cities with a population of greater than 5 million people being defined as large as detailed in the Methods section. We expect agglomeration costs to be more marked in the latter. The sample is split unequally, with 36 large cities from 18 countries, around 35% of the sample of firms. We illustrate the results for the innovation index in Fig 3, the red line being for large cities and the yellow line for small cities. As in Fig 2, to facilitate interpretations of the

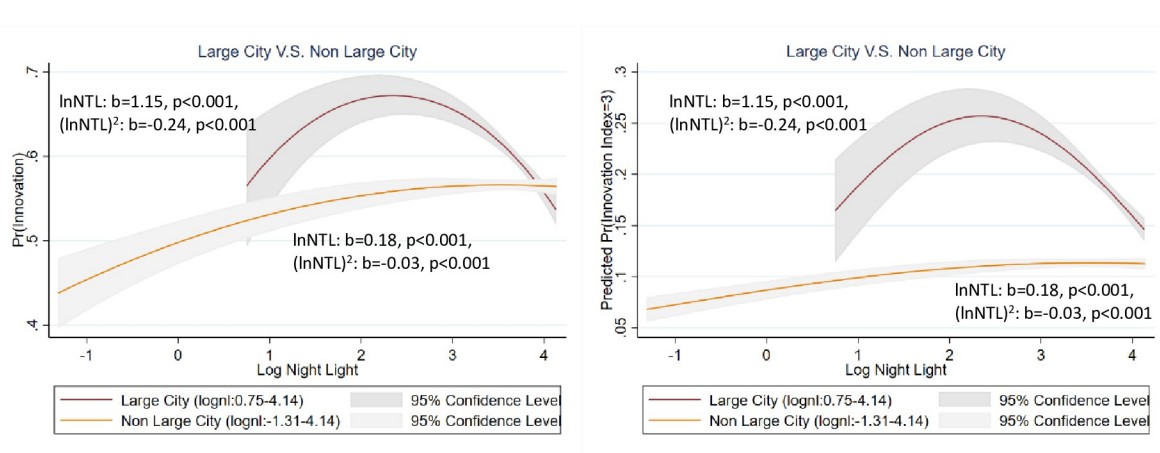

**Fig 3. Innovation and night light in large and small cities.** This Figure summarizes predicted outcomes of innovation from ordered logit models with 95% significance levels. The ordered logit models regress two innovation outcomes on a quadratic equation of logarithm of nightlight density: the outcome in **a** refers to innovation intensity, calculated as the cumulative (predicted) probability of having low, medium, and high levels of innovation; the outcome in **b** refers to innovation intensity, calculated as the predicted probability of having a high level of innovation. Confidence intervals in **b** are derived directly from the ordered logit regression. For **a**, as it combines three out of the four outcomes Pr(Innovation Index = 1, 2 and 3) in ordered logit regressions, we calculate the confidence intervals for Pr(Innovation Index = 0) from ordered logit regressions first. We then use one minus the above intervals to approximate the confidence intervals for Pr(Innovation Index = 1, 2 and 3). The results from this construction are very similar to the ones obtained directly from logit models where the dependent variable is a dummy variable based on whether the innovation index is greater than 0. In each graph, we estimate these ordered logit models for large and small cities separately. Large cities are defined as those with >5 million population and small cities as those with <5 million population. Nightlights are measured in natural logarithms and lagged. All equations include lagged GDP per capita, regional dummy variables and year fixed effects. Reported coefficients are all statistically significant at 95% levels. Full results of regressions on the respective sub-samples upon which these figures are based are reported in S4 Table. Red lines refer to large cities and yellow lines indicate small cities. The grey shaded areas are 95% confidence intervals. The point estimates of the linear and quadratic forms of the logarithm of night light density and the corresponding two-tail p-values of the estimated parameters are written next to the curves. The results for each individual component of the index are similar. In a robustness check, we also use 10 million population as the threshold for large cities because 10 million population is the threshold for "megacities" in the UN Report on World Urbanization Prospects 2018. We do not use the 10 million threshold in our main analysis because it results in a much smaller sample of giant cities: only 12 cities are identified as megacities in our sample using that measure.

effect size, we use the predicted probabilities of two innovation outcomes as the y-axis: innovation intention (**a**) and innovation intensity (**b**). The underlying ordered logit regression results are reported in S4 Table.

In both **a** and **b** of Fig 3, the innovation curves for large cities are concave and decline more steeply than the curves for small cities as expected. There is also difference in the range of NTL density for large and small cities: the curves for large cities start to the right of the rest of the sample. This implies that urban economic density, proxied by NTL density, tends to be systematically higher in more populous cities, confirming that NTL density represents a valid indicator of city level economic and social activities. Furthermore, both curves are almost everywhere higher in large cities than in small cities, suggesting that a given level of luminosity generates more innovation in large than small cities. Finally, Fig 3 shows that high agglomeration costs are more prevalent in large cities.

We illustrate the geographic dispersion of large and small cities in Fig 4 to indicate the generality of the above findings. Fig 4 plots the distribution of all large (**a**) and small cities (**b**) in our sample as well as the nightlight density and the average innovation index among all firms in representative cities which are selected to ensure maximal coverage of continents. Large cities consistently play important roles in driving innovation while suffering from large agglomeration costs in a wide variety of developing countries with very different levels of urbanisation. The figure reveals two general patterns across cities: 1) the maximum average level of innovation is higher in large cities (2.53) than in small cities (2.02); 2) among large cities in **a**, innovation levels generally decrease with NTL density. On the contrary, among small cities in **b**, innovation levels generally increase with NTL density. The latter suggests that agglomeration costs do not outweigh agglomeration benefits in small cities, but they do in large ones.

We conduct a series of robustness checks with alternative model specifications such as using OLS instead of ordered logit models and adding more firm-level control variables to show our results are not driven by heterogeneities across firms. All provide further support for our findings. See the Methods Section and Supporting Information for details.

## Discussion

This paper adds to the ongoing discussion on understanding the nature of cities in developing countries and echoes the insight that studies on urban economics in developing countries should focus more on the downside of urban density [21]. We provide consistent measures of innovation and agglomeration economies at the city level in developing countries, across a wide variety of national contexts, using WBES firm-level data matched by geo-location with NTL density data. This is, to our knowledge, the largest cross-country firm-level sample used to consider urban agglomeration effects on innovation. The cities were selected using an algorithm to ensure that the sample contained cities of comparable levels of demographic significance in each country (details are in the Methods section). On this basis, we explored whether the large net agglomeration benefits for innovation at the city level identified in the existing literature for developed economies also pertain in developing countries.

We find evidence that in cities in developing countries, agglomeration costs associated with sustainable development challenges can be large enough to offset agglomeration benefits on innovation, and this effect is strongest in the largest cities. Large cities in developing countries may therefore not be able to act as sustainable sources of innovation. Furthermore, high urban economic density sits at the root of key challenges to sustainable cities. The costs of agglomeration have previously been noted in the development economics literature and have been associated with insufficient public goods provision and overcrowding [53], a point also

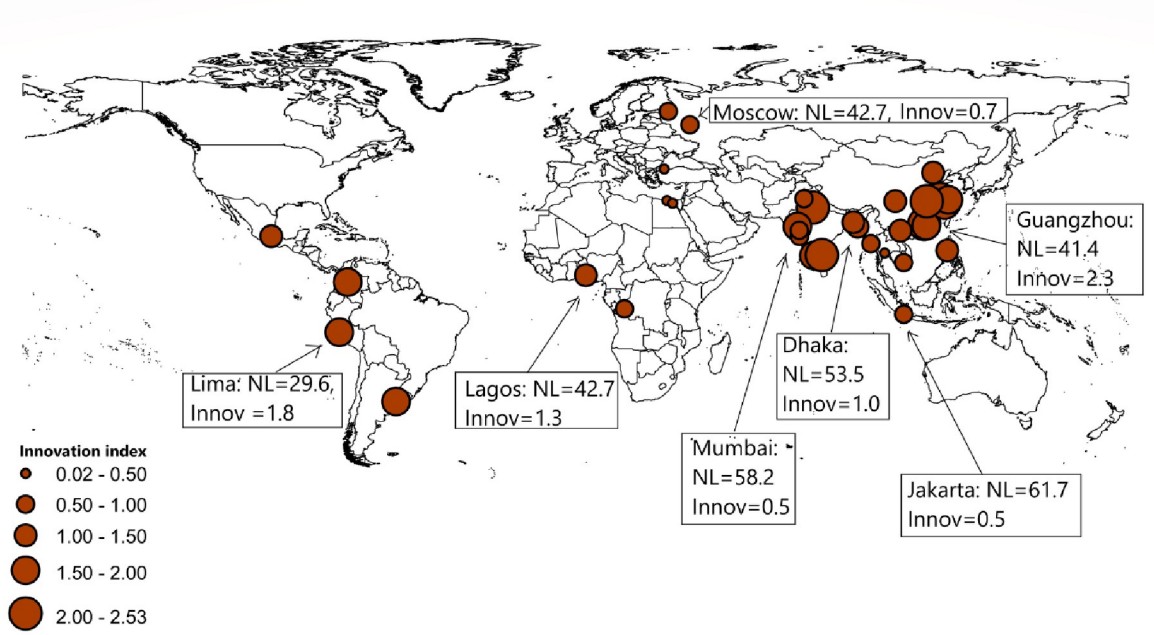

Panel a. Large cities

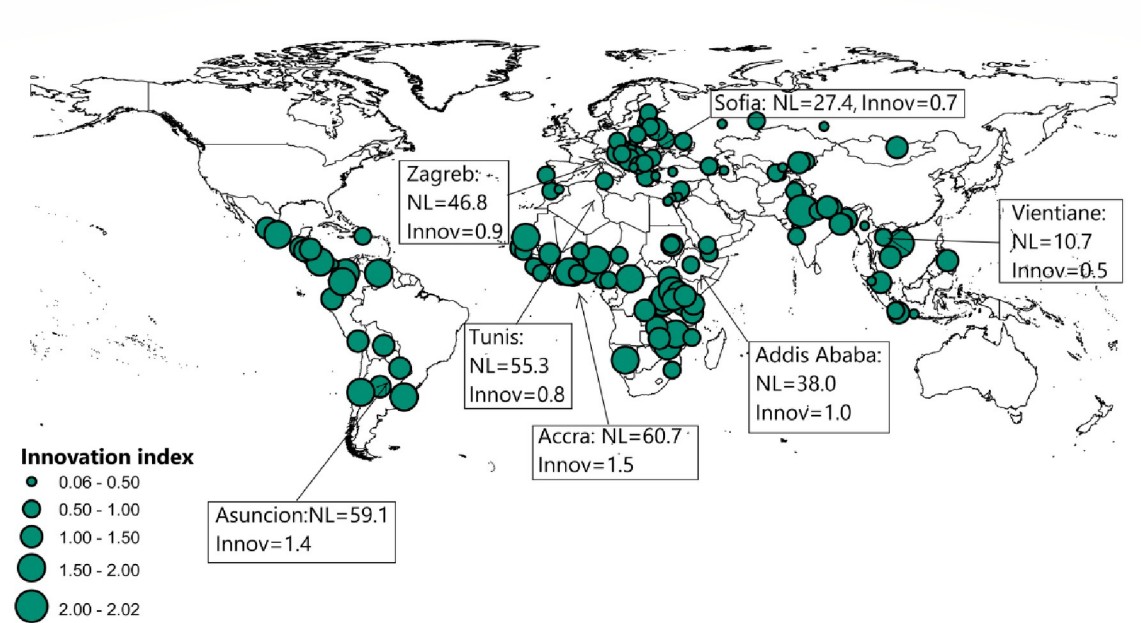

Panel b. Small cities

**Fig 4. Geographical distribution of large cities and small cities.** This Figure plots the distribution of large and small cities and the average innovation index among all firms in these cities. As before, large cities are defined as those with >5 million population. Panel **a** plots 36 cities in our sample that are defined as "large" cities and **b** plots 128 "small" cities with <5 million population. The Figure also highlights some representative cities in each category with their corresponding nightlight density and innovation levels. Representative cities are selected to ensure maximal coverage of continents. The Figure reveals two patterns: 1) the maximum level of innovation is higher in large cities (2.53) than in small cities (2.02); 2) among large cities in **a**, innovation levels in general decrease with nightlight density. On the contrary, among small cities in **b**, innovation levels in general increase with nightlight density. National/regional specificity and the level of country

development both have weak explanatory powers in predicting which cities have positive agglomeration effects and which have negative ones. Nightlight data is based on the harmonised version derived by Li et al. (2020) because different satellites were used in 1992–2013 (Defense Meteorological Satellite Program (DMSP)) and 2012–2018 (Visible Infrared Imaging Radiometer Suite (VIIRS) from the Suomi satellite). City location comes from the Database of Global Administrative Areas (https://gadm.org/download_country_v3.html).

emphasised by the United Nations (https://www.un.org/sustainabledevelopment/cities/). However, agglomeration effects have typically not been considered in the context of innovation. That is, analysis of urban agglomeration and innovation in developing countries is relatively scarce, and many of the existing studies are within one or a few countries. For example, recent literature explored the relationship between firm-level innovation and population density in a large sample of Asian countries. But the study does not have a global scale, nor does it consider the agglomeration costs and the possibility that the relationship can be concave [30]. Our firm-level analysis also provides a nuanced understanding of the mechanisms underlying urban agglomeration by uncovering how individual agents respond to urban economic density. Our results suggest two common regularities in developing countries: a) for innovative activities, agglomeration benefits are largely offset by agglomeration costs; b) large cities innovate more, but their higher agglomeration costs prevent them from achieving their full innovation potential.

These findings can deepen our understanding of the joint importance of sustainability of large cities and innovation in developing countries whose development paths have differed from those in developed countries. Thus, basing the existing theory and evidence on developed countries has led to a relative neglect of the costs associated with urban agglomeration in developing countries. Existing evidence from developed countries suggests the predominant role of agglomeration benefits: city-level agglomeration effects associated with both localised knowledge spillover effects and economies of scale enhance innovation in larger cities. However, it is in developing economies that the process of urbanisation has been especially pronounced in recent years, with inward migration and geographic expansion combining to produce an explosion of large new cities in many countries and all parts of the globe. Our study across a very large sample of countries and cities suggests that the pattern of agglomeration in developing countries may be different from what the literature has documented in developed countries. We identify that agglomeration costs can outweigh benefits in developing countries: in fact, in our sample the PoC is reached at quite a low level of luminosity (when NTL is 22.760, about the nightlight luminosity in Minsk–see Supporting Information, S2 Table). This is closely associated with a key feature of major cities in developing countries documented in the recent literature: that these cities can be disproportionally large and therefore may suffer from excessive agglomeration.

We are also able to use the WBES survey data to provide preliminary evidence to support the view that the above pattern for developing countries does not apply to cities in developed countries. Using the same selection criteria as documented in the Methods section, we identified 4 cities with 343 firms in developed countries from the WBES database: Stockholm, Naples, Rome and Tel Aviv. Employing a regression analysis on these observations using the same method we use for cities in developing countries in our main analysis, we find that urban agglomeration monotonically increases innovation in these cities in developed countries (results are available upon request from the authors), supporting the view that agglomeration costs are indeed not large enough to outweigh benefits in developed countries. In developing countries, increasing urbanisation has been continuing rapidly for decades along with significant migration from the countryside. While many enormous cities have been created, key elements of public goods, services and infrastructure including housing, water, electricity,

internet, and transport have not kept pace. This means that, though external and network effects may still enhance innovation, the costs of exchanging knowledge and increasing specialisation and complexity in research activities increase as the city becomes more densely populated. While in developed economies, where the scale of cities is containable and much more appropriate infrastructure has been put into place, it is likely that the factors driving potential congestion effects have been largely resolved and the positive externalities of scale yield the traditional agglomeration benefits in innovation.

Our results based on the sub-sample of large cities in developing countries contribute to the ongoing discussion about the critical role of large cities in sustainable development in developing countries [54] and are worth particular attention. We found that large cities with populations of more than 5 million have on average higher levels of innovation than smaller cities; they are engines of innovation in the developing world. However, the net benefits of scale are smaller, in that these larger cities show a more pronounced inverted U-shape than the smaller ones. This is consistent with the argument that agglomeration costs especially dampen the innovation potential of these large cities. This contrasts with existing evidence in developed economies which suggest that agglomeration benefits to innovation can accrue in large cities where physical capital and infrastructure are aligned with the cities' populations [7, 15]. However, most global megacities are in developing countries, where we conjecture that agglomeration costs may set in as population growth puts increasing strains on infrastructure, and lead to sharp declines in innovation as luminosity increases [7, 55].

We are able to provide some evidence in support of this conjecture. As WBES does not have appropriate measures of infrastructure accessibility for urban populations and very limited information on cities in developed countries, we instead collect external data from the Urban Indicators Database published by the UN-Habitat (https://data.unhabitat.org/pages/datasets), which provides information on both urban population and urban transport in cities in developing and developed countries. Using a one-way t-test, we compare the mean value of the proportion of urban population that has convenient access to public transport between the group of cities in developing countries and the group of cities in developed countries. S8 Table presents the t-test results, as well as details on the data and the measure. Results show that the public transport infrastructure measure is significantly higher in cities in developed countries than in developing countries, supporting our conjecture that agglomeration costs in cities in developing countries can mean that urban infrastructure is not able to keep pace with population growth. Our results are also consistent with recent studies using alternative measures of infrastructure [56].

Future research can extend this paper and help improve our understanding of urban agglomeration in developing countries in four ways. First, the cross-sectional structure of our database results in some limitations of our study. A full dynamic framework may be needed to capture the development path of cities in middle- and low-income countries that are growing rapidly [23, 57, 58]. A longitudinal dataset could extend this study by tracking the development path of these fast-growing cities and enrich our understanding of how to address large agglomeration costs over time. Moreover, we use a one-year lag in the NTL density in this paper. A longitudinal dataset could also deepen our understanding of the temporal dynamics of the innovation process and figure out the optimal time lags of urban agglomeration measures. Second, future research might consider using structural modelling approaches to explicitly document the urbanisation process in developing countries based on spatial equilibrium models [58]. This can help illustrate how the fundamentals of urban agglomeration in developing economies differ from those in developed countries. Third, future research can investigate the mechanisms of large agglomeration costs in developing countries and quantify the impact of each factor in hindering agglomeration effects on firm-level innovation: market structure,

disease, crime rates or congestion. Fourth, future research can extend beyond administrative boundaries and explore alternative approaches to define cities. For example, research can investigate the spatial reach of agglomeration economies [59, 60] by experimenting on different spatial scales ranging from the vicinity of firms to large metropolitan areas.

Finally, our study has significant business and policy implications. We know that urban agglomeration plays a critical role in innovation and growth [4, 5, 15]. Yet our findings suggest that the most obvious potential innovation engines in developing countries, large cities, suffer particularly from agglomeration costs. Consistent with the literature documenting the fundamental difference in urban systems between developing and developed countries [23], our paper also suggests that policies mostly based on the experience of developed countries might be misleading in solving problems in developing countries. Acknowledging the feature of excessive agglomeration in developing countries, we propose three ways to address the problem of agglomeration costs in large cities in developing countries. First, both domestic and international firms may reconsider where they locate their innovative activities, because in developing countries the greatest agglomeration benefits are not always associated with the largest urban populations. Our results suggest that firms seeking external benefits in innovation might choose to locate in smaller cities which are not yet innovation hubs but may still provide higher net agglomeration benefits. In addition, state policies to encourage innovation might sensibly focus on those relatively smaller cities in which the spillover benefits of innovation outweigh agglomeration costs. Second, policymakers in developing countries should address urban problems in large cities such as weak infrastructure and high levels of congestion. This echoes the call for making cities in developing countries "more livable" [61], reducing the negative externalities in cities and improving urban quality of life [34]. Third, policymakers might consider facilitating collaborations between large and small cities to explore their comparative advantages. This can be done through promoting inter-regional linkages [62], building ecosystems [32], and in particular developing city-regions [63] where large cities, as innovation hubs, are connected with their surrounding relatively smaller cities whose spillover benefits of innovation can outweigh agglomeration costs.

## Methods

### Data

To extend the analysis of city-level agglomeration effects on firm innovation from developed to developing countries requires selecting representative cities and identifying their boundaries. Our selection of cities balances the trade-off between a) having enough variation in population sizes across cities for reliable statistical inference and b) ensuring that, within a cross-country setting, we select cities of similar importance in countries of different sizes. Moreover, the literature suggests that the spatial inequality of cities in developing countries is high: a large proportion of population and economic activities tends to concentrate in few major cities [23]. In building our sample of country-cities, we select fewer (more) cities in smaller (larger) countries, focusing on cities where agglomeration effects are likely to exist, and cities that are of comparable importance given their country context. We use these criteria because the distribution of cities in small countries is likely to follow the law of the Primate City where population concentrates in the largest cities. For example, in Costa Rica, the relative primacy (the ratio of the primate city's population to that of the second largest city) of San Jose is larger than 35. Large countries have a more even distribution of city sizes, following Zipf's law where several cities jointly play dominant roles. For example, in Bangladesh, the relative primacy of Dhaka is smaller than 4. In fact, our data show a negative relationship between the rank of the percentage of population in the largest city and the rank of country population size.

Our selection method is to divide countries into categories by size and select different numbers of large cities in rank order of population in different countries. Our classification of countries and the choice of the cut-off values are based on structural break tests on the distribution of country population size. Of the 115 countries originally in the WBES sample and excluding China and India as extreme outliers with populations more than three times as large as any other country, we use the Bai-Perron test, which is designed for multiple unknown structural breaks, to identify four structural breakpoints in the distribution of country population size. We obtain cut-off values of 20 million, 50 million, 100 million, 200 million people. We therefore select 1, 2, 3, 5, 6 and 12 cities from countries whose 2019 population sizes are: <20 million, 20–50 million, 50–100 million, 100–200 million, 200 million-1 billion, >1 billion (this category includes only China and India). The country-city distribution is mapped in Fig 1, Panel **a**.

Urban areas can be defined by either administrative boundaries or economic characteristics that may not be aligned with administrative units. We define cities in our sample based on administrative units for three reasons. First, the UN Report on 2018 World Urbanization Prospects (https://population.un.org/wup/Publications/Files/WUP2018-Report.pdf) proposed four approaches to define urban areas but predominately relied on administrative criteria. A meta-analysis on 70 studies covering 21 developing and 12 advanced countries also confirmed that urban agglomeration is measured at geographical units equivalent to administrative levels 2 or 3 (municipalities or districts) in most studies [53]. Second, planning and organisation of infrastructure, which are key to agglomeration benefits and costs, are conceptually based on administrative units. Third, it is difficult to extend definitions of cities that do not rely on administrative units to a global context, because these definitions involve selecting thresholds that are specific to certain countries. For example, a larger population threshold should be selected for Asian countries and a smaller land area threshold might be selected for European countries. It might also be hard to know what these thresholds should be. In comparison, a definition of cities based on administrative units respect the differences in urban development across countries because administrative subdivisions are integrated economic and social units in each country.

One concern with administratively defined cities is that they may contain sparsely populated rural areas. To alleviate this concern, we define city territories by combining polygons of administrative subdivisions covering mostly urban and well-developed suburban areas within each administrative city and excluding rural subdivisions. Maps indicating the boundaries of administrative subdivisions are obtained from the Database of Global Administrative Areas (https://gadm.org/download_country_v3.html), a database that has the global coverage for administrative subdivisions and is explored by studies on urbanisation in top journals [64]. Our administrative cities are defined at administrative levels 2 or 3 (districts for capital cities or large urban areas, and municipalities for others), following mainstream studies [53].

Our firm level information is derived from the World Bank Enterprise Survey data (WBES). The World Bank has since 2006 undertaken firm-level surveys mostly in developing and emerging economies using a standard survey instrument. Each survey is a global stratified random sample, with strata chosen to reflect variation in firm size, business sector, and geographic region to facilitate cross-country comparisons. These data are increasingly used in the social sciences [65–67]. In S1 Fig, we present the distribution of firm sizes in WBES to verify that the firms selected by WBES encompass a representative sample of businesses in developing countries. In our sample, 43% of firms are small, 35% are medium and 22% are large, suggesting that WBES did not predominantly focus on large-scale businesses. Meanwhile, the World Bank provides geo-positioning data since 2011 which allow us to locate each surveyed firm within a city by matching the GPS coordinates of firms with the city territories we define.

Furthermore, the survey contains firm-specific information on innovative activity, including whether the firm introduced new products, new processes, and/or undertook R&D activity. The WBES data therefore provides distinct measures of innovation activities within each firm, following the OECD (2005) definition of innovation distinguishing between new products and processes, as well as considering R&D activity.

In matching cities to WBES, we exclude very small cities (with fewer than 250,000 population) from the sample, consistent with the UN definition of small cities [28]. This also excludes ghost cities and ensures that our NTL measure captures actual human activities. Administrative cities with small populations often do not display the agglomeration characteristics which we study, and borders for small cities are often quite vague. We also exclude cities with fewer than 20 firms surveyed by the World Bank because valid statistical inferences require a reasonably large sample size of firms in each city. We then applied two further exclusion criteria. First, because our proposition applies to developing countries but not to developed ones, we exclude countries if they were relatively highly developed. We formalised this criterion in terms of GDP per capita in excess of $30,000. Second, war or civil strife, which can be measured at a local as well as national level, are a regular feature in some developing countries and may disrupt the relationship between urban economic activity and innovation. We therefore exclude all large, localised conflicts which occurred in the year of the WBES survey. Following the Uppsala Conflict Data Programme (UCDP), a large conflict entails more than 1000 battle deaths per year [68]. The conflict data provides GPS coordinates for the location of each battle in each year. We select the ones with >1000 deaths per year and match their location with the city territories. A city is excluded from our sample if any of those battles occurs in the year of our sample and falls within the city. Using these criteria, we exclude 5 cities from the survey as conflict zones. Table 1.1 in S1 Table reports how these exclusion criteria affect the sample size. Thus, the raw WBES dataset covers 115 countries and more than 111,000 firms but once we focus only on cities of similar importance in different countries, the sample becomes 103 countries containing 186 cities, with 37,259 firms. These exclusion criteria, together with excluding observations where night light does not reflect economic activities due to gas flare, as well as observations with missing data on GDP and all innovation measures, reduce the sample to 96 countries, 164 cities and a maximum of 34,690 firms depending on the innovation measure. Notes to Table 1.1 in S1 Table detail the impact of each exclusion.

Table 1.2 in S1 Table shows the representativeness of our sample after the selection process in Table 1.1 in S1 Table. By selecting different numbers of cities in countries with different population sizes and by implementing the above exclusion criteria, we face a trade-off between focusing on cities that are of comparable importance across countries and getting a representative sample of all firms surveyed in WBES. In Table 1.2 in S1 Table, we calculate the proportion of firms in our sample over all firms surveyed in WBES by categories of countries. Table 1.2 in S1 Table suggests no systematic selection bias because neither small nor large countries are oversampled. Similar proportions of firms are included in our sample in different categories of countries except in countries with 3 cities selected (over-represented) and countries with 6 cities selected (under-represented).

We use NTL data to measure urban economic density, consistent with the literature [23]. Nightlight density is measured as the average light luminosity within a city boundary defined above, ranging from a value of 0 to 63. The NTL data are lagged for one year to alleviate potential problems of reverse causality. We use the harmonised version of nightlight data derived by Li et al. (2020) [69] because different satellites were used in 1992–2013 (Defense Meteorological Satellite Program (DMSP)) and 2012–2018 (Visible Infrared Imaging Radiometer Suite (VIIRS) from the Suomi satellite). The harmonised version reconciles the inconsistency between DMSP and VIIRS while preserving the advantages of VIIRS. Noises

from gas flare, aurora, fires and other temporal lights were also removed. However, the harmonised version may suffer from the problem of top-coding. As the maximum value of NTL is restricted to 63, all very bright urban centres may have the same NTL value of 63, even though there are variations in their actual NTL luminosity. This top-coding problem does not impose serious concern over our results, because only 5 cities are at the NTL value of 63 in S2 Table. More importantly, as described in the "Results" section, our PoC occurs when NTL is 22.760, suggesting the declining part of the inverted-U curve is not purely driven by the NTL value of 63. Our arguments are also consistent with the literature which documents that top-coding is not a serious problem in most places in developing countries [70–72]. We illustrate in S2 Fig the heterogeneity in luminosity across our sample by comparing cities with low, moderate, and high levels of luminosity as well as the location of firms surveyed by WBES within these city zones. The figure shows the distribution of firms in five cities ranked from the lowest through the median to among the highest light density: Juba (South Sudan); Bogota (Columbia); Ankara (Turkey); Moscow (Russia) and Kolkata (India). The respective night light densities (ascending rank order in parentheses) are 0.3 (1); 22.9 (58); 38.1(114); 47.8 (148); and 63 (228). For each city, the number of firms shown in the figure are 402; 912; 231; 353; and 261. The full list of cities, number of firms and NTL densities are reported in S2 Table.

The World Bank repeated its surveys in some countries but surveyed different firms in different years. Thus, it can be seen in S2 Table that 66 cities are sampled twice. However, the WBES data are at the firm level and while the city may sometimes be the same (at a different date), the firms in each sample are not. Our analysis is of different firms across cities and years, so we included repeat samples but control for this in our estimated equations by including a dummy variable for the year of the survey. We therefore match NTL luminosity data with WBES based on cities and survey years in S2 Table. For example, if a city was surveyed in 2018 in WBES, we match its 2017 NTL density (taking a one-year lag) with the innovation outcomes of firms in this city in 2018.

In our regression analysis, we explore the relationship between NTL density in a city and an index of innovation activities in each firm. The construction of the index is discussed in the Introduction. The WBES data has some missing values in the measures of innovative activity, and these slightly affect the final sample: there are four cities in two countries where all the innovation measures are missing. Hence, the sample on which we estimate covers 96 countries and 164 cities. Moreover, the individual innovation measures are occasionally missing for some firms. The final sample size for the innovation index is 31,798 firms but is larger for each individual component.

We illustrate the distribution of the innovation index across cities in our sample in Fig 1, Panel **b**. This shows the average value of the innovation index across firms in each city ranges from zero to more than 2.5. The cities are of course located within our sample countries, which vary by the level of development. We have grouped countries into five categories of level of development by GDP per capita, ranging from around $245 to almost $25,000. Fig 1, Panel **b** does not suggest any monotonic relationship between average innovation activity in each city and GDP per capita of the country.

**Specification and estimation.** We estimate equations linking our measure(s) of innovation at the firm level in each city and the density of night lights in that city. We explored a variety of specifications of NTL, namely in levels and quadratic form, in log and quadratic log form. Our final and preferred regression model estimates a quadratic polynomial function of NTL. We use the logarithm of NTL density instead of levels because the logarithmic specification results in a higher value of goodness of fit, but results are robust to the use of night light in levels.

Our firm-level, cross-city, cross-country, cross-time dataset raises several dimensions of unobserved heterogeneity which we address in our empirical work. Because the dataset is of developing economies, at the country level we used a variety of standard controls, notably the level of development (country-level GDP per capita) included in lagged form to address potential reverse causality. A potentially more rigorous control for unobserved cross-country heterogeneity would be to use country fixed effects. However, because the sample includes small countries surveyed once with only one city selected (30 out of 229 country-city-year units), country fixed effects will not absorb much of the variation in city-level NTL. Moreover, the widespread use of dummy variables in country fixed effects may lead to the problem of data separation, which can result in non-convergence of maximum likelihood functions [73]. To resolve these potential problems while still controlling for some elements of unobserved cross-country heterogeneity, we combine countries into aggregate regional groups and control for regional fixed effects. Based on the World Bank methodology (https://datahelpdesk. worldbank.org/knowledgebase/articles/906519-world-bank-country-and-lending-groups), the countries in our sample are grouped into six geographic regions: East Asia & Pacific, Europe & Central Asia, Latin America & Caribbean, Middle East & North Africa, South Asia, and Sub-Saharan Africa. To address heterogeneity caused by macro-economic factors such as business cycles, we also control for the year in which the sample was taken.

Given the dependent variable takes the ordinal categorical values of 0, 1, 2 and 3, we applied ordered logistic regression methods for the main results [74]. Discussion of the assumptions of the ordered logit model is contained in S1 File. Suppose $Y_{ijt}$ is the composite innovation index for firm $i$ in city $j$ surveyed in year $t$, taking values of 0, 1, 2 and 3. $Y_{ijt}^*$ is the underlying continuous latent variable with thresholds $(k_1, k_2, k_3)$ that determine the values of $Y_{ijt}$. $NTL_{jt}$ is the city-level NTL density with one-year lag. $Z_{ijt}$ include all control variables and fixed effects with one-year lag for time-varying variables. The ordered logistic regression model is specified as:

$$Y_{ijt}^* = a + \beta_1 * NTL_{jt-1} + \beta_2 * NTL_{jt-1}^2 + \delta * Z_{ijt-1} + \epsilon_{ijt}$$

$$Y_{ijt} = \begin{cases} 0, & if\ Y_{ijt}^* \leq k_1 \\ 1, & if\ k_1 < Y_{ijt}^* \leq k_2 \\ 2, & if\ k_2 < Y_{ijt}^* \leq k_3 \\ 3, & if\ Y_{ijt}^* > k_3 \end{cases}$$

However, our results are robust to alternative model specifications. In addition to the composite innovation index, we fit standard logistic regression models for individual innovation measures (binary variables) in columns 1–3 in Table 1. OLS models in S5 Table also generate similar results to the ordered logit models. For independent variables, in S6 Table we include two additional firm-level controls using the WBES data: firm size (a dummy on whether a firm is of medium size with 20–99 employees and a dummy on whether a firm is of large size with 100 or above employees), and firm age (difference between the survey year and the year of establishment). Results in S6 Table are similar to the main results in Table 2. In unreported regressions, we also estimate the baseline model in a more parsimonious form by excluding the regional fixed effects and GDP per capita. In addition, we estimate the baseline model using country fixed effects rather than the regional groupings. We identify the same statistically significant concave relationship between night light density and innovation in all these regressions. Our results are also robust to alternative exclusion rules when we add countries in war zones in our sample.

Our conjecture of higher agglomeration costs in heavily populated cities implies congestion effects will be more marked in cities with larger population sizes. We consider the impact of city population on the innovation-night light relationship by adding an interaction term between city population size and night light density in our ordered logit regression. Both Fig 2 and the underlying regression results in S3 Table suggest that the declining part of the innovation-urban density relationship is driven by cities with large population sizes. This is further verified by our split-sample estimates based on city population. Following the definition of giant cities in the UN Report on 2018 World Urbanization Prospects, we identify cities with a population of greater than 5 million people as "large" (36 out of 164 cities in our sample). We split our city sample into large (>5 million) and small (< = 5 million) cities and run ordered logit regressions on the respective sub-samples. Fig 3 and the corresponding regression results in S4 Table are consistent with the pattern in Fig 2 and S3 Table. Data on city population size is obtained from the most recent census data for each country, and only counts the population in the city proper or urban areas, whichever is more consistent with the city boundary defined in our analysis. In an unreported robustness check, we also used 10 million population as the threshold and obtained similar results.

## Supporting information

**S1 Table. Sample descriptions.** This table has two separate parts: S1.1 and S1.2. Each has its own notes.
(DOCX)

**S2 Table. Data description.** The number of firms in each city is obtained by matching firms' GPS coordinates with city boundaries. Cities that have fewer than 20 firms are excluded from our sample, including: Sofia in 2019, Bekasi, Seberang Perai, Ecatepec, Benin City, Kazan in 2019, Toamasina, Tianjin, Basrah, Fez in 2013, Krakov in 2013, Samarqand in 2013, Rajshahi. Sofia, Kazan, Fez, Krakov and Samarqand. The total number of cities in the list is larger than the total number of distinct cities surveyed in WBES because some cities were surveyed more than once in different years and they were double counted because different firms were surveyed in different years.
(DOCX)

**S3 Table. Regressions for Fig 2: Moderating effects of city population size.** The table reports the ordered logit regression for the innovation index. We control for per capita GDP in each country and include geographic region and year fixed effects. The key independent variables are the interaction between city population and nightlight density, and the interaction between city population and the quadratic term of nightlight density. P-values are in parentheses, and 95% confidence intervals are in square brackets below p-values. Ordered Logit estimates do not include a constant. NTL and GDP are lagged. City population data comes from the most recent census data in each country. *** p<0.01, ** p<0.05, * p<0.1.
(DOCX)

**S4 Table. Regressions for Fig 3: Split sample for large and small cities.** The table reports ordered logit regression for the innovation index for large cities (>5M population) in column 1 and small cities (<5M population) in column 2. The key independent variables are the night-light density and its quadratic term. We control for per capita GDP in each country and include geographic region and year fixed effects. P-values are in parentheses, and 95% confidence intervals are in square brackets below p-values. Ordered Logit estimates do not include a constant. NTL and GDP are lagged. Coefficients can be interpreted as the increase in the log odds of being in a higher innovation level versus all lower innovation levels after a 1% increase

in nightlight density. *** p<0.01, ** p<0.05, * p<0.1.
(DOCX)

**S5 Table. Robustness check: Using OLS regressions.** The table reports OLS regression results for all previous results based on ordered logit models. Columns 1–4 report OLS results for the corresponding columns in Table 2. Column 5 relates to the model in Table in S3 Table. Columns 6 and 7 report the corresponding OLS results based on ordered logit models in Table in S4 Table. We control for per capita GDP in each country and include geographic region and year fixed effects. P-values are in parentheses, and 95% confidence intervals are in square brackets below p-values. NTL and GDP are lagged. *** p<0.01, ** p<0.05, * p<0.1.
(DOCX)

**S6 Table. Robustness tests: Adding firm-level controls.** The table reports, with firm level controls, logit regressions for each individual item in the innovation index in columns 1–3, and the ordered logit regression for the innovation index in column 4. The key independent variables are the logarithms of nightlight density and its quadratic term. We control for per capita GDP in each country and include geographic region and year fixed effects. We additionally control for firm-level controls, i.e., age, firm size dummies indicating whether a firm is medium-sized (20–99 employees) and whether a firm is large-sized (100 and over 100 employees). P-values are in parentheses, and 95% confidence intervals are in square brackets below p-values. Ordered Logit estimates do not include a constant. Number of observations varies because of missing values for each measure. NTL and GDP are lagged. Coefficients can be interpreted as the increase in the log odds of being in a higher innovation level versus all lower innovation levels after a 1% increase in nightlight density. *** p<0.01, ** p<0.05, * p<0.1.
(DOCX)

**S7 Table. Generalised ordered logit model: Testing the assumption of ordered logit model.** The table reports generalized ordered logit regressions which relax the proportional odds assumption of standard ordered logit regressions. Columns 1, 2 and 3 present the coefficients for the likelihood of: having zero versus all other categories of innovation; having zero or low innovation versus medium and high innovation; having zero, low and medium innovation versus high innovation. The key independent variables are night light density and its quadratic term. We control for per capita GDP in each country and include geographic region and year fixed effects. P-values are in parentheses, and 95% confidence intervals are in square brackets below p-values. Number of observations varies because of missing values for each measure. NTL and GDP are lagged. *** p<0.01, ** p<0.05, * p<0.1.
(DOCX)

**S8 Table. Urban transport infrastructure in large cities in developed and developing countries: T-test results.** Data comes from the Urban Indicators Database published by the UN-Habitat (https://data.unhabitat.org/pages/datasets), which covers information on both urban population and urban transport in cities in developing countries as well as developed countries. We first identify large cities as those with more than 5 million urban population in 2015, consistent with our main analysis, which leads to 45 cities in developing countries and 14 in developed countries. The Urban Indicators Database provides information on the proportion of urban population that has convenient access to public transport, defined as the estimated share of urban population with access to a public transport stop within a walking distance of 500 meters (for low-capacity public transport systems) and/or 1000 meters (for high-capacity public transport systems). We perform a one-way t-test to compare the mean value of the proportion of urban population that has convenient access to public transport

between 45 cities in developing countries and 14 cities in developed countries. The table shows the mean values, standard deviations and confidence intervals for the two groups, and the t-statistics and p-value for the difference in their mean values.
(DOCX)

**S1 Fig. Distribution of firm sizes in our sample.** The Figure shows the distribution of firm sizes in our sample. The WBES surveys distinguish firms into small firms (5–19 employees), medium firms (20–99 employees), and large firms (100+ employees). The figure shows that in our sample, 43% of firms are small, 35% of firms are medium firms and 22% are large firms, suggesting that our sample is representative with respect to firm sizes.
(TIF)

**S2 Fig. Nightlight densities in the lowest, around the median and the highest ranked cities in the sample and the distribution of firms in each city.** The Figure shows the distribution of firms in five cities ranked from the lowest through the median to among the highest light density: Juba (South Sudan); Bogota (Columbia); Ankara (Turkey); Moscow (Russia) and Kolkata (India). The respective night light densities (ascending rank order in parentheses) are 0.3 (1); 22.9 (58); 38.1(114); 47.8 (148); and 63 (228). For each city, the number of firms shown in the Figure are 402; 912; 231; 353; and 261. The number 228 is larger than the total number of cities because some cities were surveyed more than once in different years and are double counted because different firms were surveyed in different years. Night light data is based on the harmonised version derived by Li et al. (2020) because different satellites were used in 1992–2013 (Defense Meteorological Satellite Program (DMSP)) and 2012–2018 (Visible Infrared Imaging Radiometer Suite (VIIRS) from the Suomi satellite). Darker colour indicates higher night light luminosity. Maps indicating the boundaries of urban and suburban administrative subdivisions in each city are obtained from the Database of Global Administrative Areas (https://gadm.org/download_country_v3.html). The Pearson correlation between the city rank by luminosity and GDP per capita is 0.24, which partly confirms that our NTL measure is a good proxy for social and economic activities. Each dot refers to a firm. The location of the firm is based on the GPS coordinate in WBES. Source: own calculations.
(TIF)

**S1 File. Testing the assumption of the ordered logit model.**
(DOCX)

## Author Contributions

**Conceptualization:** Saul Estrin, Yuan Hu, Daniel Shapiro, Peng Zhang.

**Data curation:** Yuan Hu, Peng Zhang.

**Formal analysis:** Yuan Hu, Peng Zhang.

**Funding acquisition:** Peng Zhang.

**Methodology:** Saul Estrin, Yuan Hu, Daniel Shapiro, Peng Zhang.

**Validation:** Saul Estrin, Daniel Shapiro, Peng Zhang.

**Visualization:** Yuan Hu, Peng Zhang.

**Writing – original draft:** Saul Estrin.

**Writing – review & editing:** Saul Estrin, Yuan Hu, Daniel Shapiro, Peng Zhang.

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
