## [Decision Letter · Decision Letter 0]

23 May 2024

PONE-D-24-17021Urban agglomeration and innovation in developing economiesPLOS ONE

Dear Dr. Zhang,

Thank you for submitting your manuscript to PLOS ONE. After careful consideration, we feel that it has merit but does not fully meet PLOS ONE’s publication criteria as it currently stands. Therefore, we invite you to submit a revised version of the manuscript that addresses the points raised during the review process.

We look forward to receiving your revised manuscript.

Kind regards,

Gang Xu, Ph.D.

Academic Editor

PLOS ONE

Journal Requirements:

2. For studies involving third-party data, we encourage authors to share any data specific to their analyses that they can legally distribute. PLOS recognizes, however, that authors may be using third-party data they do not have the rights to share. When third-party data cannot be publicly shared, authors must provide all information necessary for interested researchers to apply to gain access to the data. (https://journals.plos.org/plosone/s/data-availability#loc-acceptable-data-access-restrictions) 

3. We note that Figures 1, 4 and Supporting Figure 2 in your submission contain map images which may be copyrighted. All PLOS content is published under the Creative Commons Attribution License (CC BY 4.0), which means that the manuscript, images, and Supporting Information files will be freely available online, and any third party is permitted to access, download, copy, distribute, and use these materials in any way, even commercially, with proper attribution. For these reasons, we cannot publish previously copyrighted maps or satellite images created using proprietary data, such as Google software (Google Maps, Street View, and Earth). For more information, see our copyright guidelines: http://journals.plos.org/plosone/s/licenses-and-copyright.

1) You may seek permission from the original copyright holder of Figures 1, 4 and Supporting Figure 2 to publish the content specifically under the CC BY 4.0 license.  

2) If you are unable to obtain permission from the original copyright holder to publish these figures under the CC BY 4.0 license or if the copyright holder’s requirements are incompatible with the CC BY 4.0 license, please either i) remove the figure or ii) supply a replacement figure that complies with the CC BY 4.0 license. Please check copyright information on all replacement figures and update the figure caption with source information. If applicable, please specify in the figure caption text when a figure is similar but not identical to the original image and is therefore for illustrative purposes only.

**Additional Editor Comments:**

Please revise your paper following comments from the two reviewers. All comments should be addressed carefully. Detailed responses should be uploaded point-by-point.

Reviewers' comments:

Reviewer's Responses to Questions

**Comments to the Author**

1. Is the manuscript technically sound, and do the data support the conclusions?

Reviewer #1: Yes

Reviewer #2: Partly

2. Has the statistical analysis been performed appropriately and rigorously? 

Reviewer #1: Yes

Reviewer #2: Yes

3. Have the authors made all data underlying the findings in their manuscript fully available?

Reviewer #1: Yes

Reviewer #2: Yes

4. Is the manuscript presented in an intelligible fashion and written in standard English?

Reviewer #1: Yes

Reviewer #2: Yes

5. Review Comments to the Author

Reviewer #1: The paper "Urban Agglomeration and Innovation in Developing Economies" presents two significant contributions to the field of urban economics and innovation studies. Firstly, the authors introduce a novel framework that challenges the conventional understanding of the relationship between urban agglomeration and innovation in developing countries. This framework suggests that urban agglomeration may not lead to sustainable improvements in innovation within these contexts, contrary to prevailing assumptions. This conceptual shift represents a significant contribution to the discourse on urban development and innovation in developing economies. Secondly, the paper addresses the methodological challenges faced by empirical studies on a global scale, particularly in developing countries. The authors highlight the difficulty of obtaining comprehensive and consistently defined measures, as well as the unreliable collection of data related to urban agglomeration and innovation. To overcome these challenges, the authors leverage recent advancements in data collection techniques to create a unique database that measures both variables consistently across a large sample of developing countries. This methodological approach represents a valuable contribution to the field, providing a framework for future research in similar contexts.

Here are two advices:

1. While the paper highlights the importance of the dataset, there are areas where the presentation could be improved. The authors extensively discuss the selection process of cities in the dataset, but they could provide more innovative insights into how the new data is integrated. It would be beneficial for the paper to clearly demonstrate how the dataset is reconfigured or reorganized, rather than simply being used in its raw form. Strengthening this aspect of the paper would enhance its contribution to the field.

2. The paper could benefit from improvements in readability and presentation. For instance, the introduction of the dataset could be enhanced by providing a more detailed explanation of the variables, particularly the innovation level variable. This would help non-specialist readers better understand the significance of the data and its implications. Additionally, while Table 1 presents descriptive statistics, including mean values, it would be helpful to include explanations or context for these statistics to aid in understanding their significance.

In conclusion, "Urban Agglomeration and Innovation in Developing Economies" makes important contributions to the understanding of urban agglomeration and innovation. Enhancing the presentation of the dataset, improving the readability of the paper, and providing more detailed information on the composition of innovation levels would further strengthen its impact and appeal to a wider audience.

Reviewer #2: This paper attempts to investigate impacts of the costs of excessive agglomeration on innovation in developing economies. Nightlight density is selected to proxy urban economic density and innovation data at the firm level across the developing world is acquired from the WBES. A regression model was proposed to test the relationship between innovation at the firm level and NTL density in the city in which the firm was located, and an interaction term between city population size and NTL density was added to reveal impacts of population on the innovation-NTL relationship. This paper conducted much data analysis, but some crucial questions should be addressed in a major revision.

Major:

1. The title of this paper is “Urban agglomeration and innovation in developing economies”. However, the content is mainly about the relationship between agglomeration costs and urban innovation in developing countries. The scientific question studied and the main findings obtained should be reflected in the title.

2. In the analysis of experimental results, the failure of urban infrastructure in developing countries to keep pace with population growth has been cited as a reason why large cities in developing countries are not sustainable sources of innovation. The core point is that the conjecture lacks data to support it. It is necessary to collect data related to the infrastructure construction of large cities in developing countries to analyze whether they are really unable to meet the needs of urban residents. At the same time, infrastructure data of cities in developed countries can also be collected to confirm that big cities in developed countries do not have the problem of lack of infrastructure.

3. Although the use of NTL as an indicator of urban agglomeration costs is theoretically reasonable and innovative, this research method is only applied to developing countries in this paper. Are there studies in developed countries that also use NTL to assess the impact of agglomeration costs on innovation? Are the conclusions about the differences between developed and developing countries influenced by the research method? It is suggested that cities in developed countries should be selected for comparative analysis with the same method.

Minor

Title. Three principles should be followed for the title: Accurate, Brief, and Clear. A successful title usually states a study's main findings and core conclusions. Please avoid using unnecessary words, such as “A study of/on, Investigation of, Observation on”, etc. I suggest revising your title to “Cost of excessive agglomeration limits sustainable innovation in developing economies”.

6. Abstract: The abstract is a summary of the content of your paper. It is a guide to the most important parts of your study, and it has to be able to stand alone. I recommend writing the abstract following the guideline of the Nature publisher (https://www.nature.com/documents/nature-summary-paragraph.pdf). For your paper, abstract can be a more comprehensive summary of the full text. In addition, please extend the abstract in one paragraph with 200-250 words.

7. Keywords. Keywords are a tool to help readers and search engines find your paper. Keywords should contain words and phrases that suggest what the topic is about (the first one or two words). Also include words and phrases that are closely related to your topic (the rest two or three words). Among your keywords, “agglomeration benefits and costs”, is an uncommon phrase, which could be replaced or revised.

1. Figure 1. The size of the thematic map and the transparency of the dots should be adjusted to make the information clearer.

2. Figure 2. The font on the axis should be larger.

6. PLOS authors have the option to publish the peer review history of their article (what does this mean?). If published, this will include your full peer review and any attached files.

Reviewer #1: No

Reviewer #2: No

---

## [Author Response · Author response to Decision Letter 0]

5 Jul 2024

We have uploaded a file, labeled as "Response to Reviewers", to address all the comments from the reviewers one by one. This letter also includes our response to the requests from the journal office. In the cover letter to the editor, we summarize the documents we submit for this round of revision, together with our response to the editor's requests.

---

## [Decision Letter · Decision Letter 1]

30 Jul 2024

Agglomeration costs limit sustainable innovation in cities in developing economies (previous title: Urban agglomeration and innovation in developing economies)

PONE-D-24-17021R1

Dear Dr. Zhang,

We’re pleased to inform you that your manuscript has been judged scientifically suitable for publication and will be formally accepted for publication once it meets all outstanding technical requirements.

Kind regards,

Gang Xu, Ph.D.

Academic Editor

PLOS ONE

Additional Editor Comments (optional):

Thank you for the revision. I recommend accepting this paper after addressing the minor issue raised by the Reviewer 1, who suggested reducing words in the Introduction section.

Reviewers' comments:

Reviewer's Responses to Questions

**Comments to the Author**

1. If the authors have adequately addressed your comments raised in a previous round of review and you feel that this manuscript is now acceptable for publication, you may indicate that here to bypass the “Comments to the Author” section, enter your conflict of interest statement in the “Confidential to Editor” section, and submit your "Accept" recommendation.

Reviewer #1: All comments have been addressed

Reviewer #2: All comments have been addressed

2. Is the manuscript technically sound, and do the data support the conclusions?

Reviewer #1: Yes

Reviewer #2: Yes

3. Has the statistical analysis been performed appropriately and rigorously? 

Reviewer #1: Yes

Reviewer #2: Yes

4. Have the authors made all data underlying the findings in their manuscript fully available?

Reviewer #1: Yes

Reviewer #2: Yes

5. Is the manuscript presented in an intelligible fashion and written in standard English?

Reviewer #1: Yes

Reviewer #2: Yes

6. Review Comments to the Author

Reviewer #1: The authors The authors have effectively addressed the concerns raised in the first round of reviews. The revised manuscript shows a clear effort to incorporate the feedback provided, particularly in refining the methodological approach and clarifying the theoretical framework. After reading the revised version, Here is only one suggestion about the introduction. I hope will help further improve the quality and readability of this article.

The introduction provides a strong rationale for the study, but it is somewhat lengthy. Consider condensing the background information to focus more sharply on the research question and objectives. for example, the literature review is comprehensive but could be streamlined. Some references to general urbanization theories could be reduced to avoid redundancy.

Reviewer #2: The author responded to each question well in the revision. The revised manuscript is rigorously structured and innovative. I consider this article suitable for acceptance and publication.

7. PLOS authors have the option to publish the peer review history of their article (what does this mean?). If published, this will include your full peer review and any attached files.

Reviewer #1: No

Reviewer #2: No

---

## [Editor Report · Acceptance letter]

5 Nov 2024

PONE-D-24-17021R1 

PLOS ONE

Dear Dr. Estrin, 

I'm pleased to inform you that your manuscript has been deemed suitable for publication in PLOS ONE. Congratulations! Your manuscript is now being handed over to our production team.

Kind regards, 

on behalf of

Dr. Gang Xu 

Academic Editor

PLOS ONE